# ATANN3 Is Involved in Extracellular ATP-Regulated Auxin Distribution in *Arabidopsis thaliana* Seedlings

**DOI:** 10.3390/plants12020330

**Published:** 2023-01-10

**Authors:** Jiawei Xu, Lijuan Han, Shuyan Xia, Ruojia Zhu, Erfang Kang, Zhonglin Shang

**Affiliations:** Ministry of Education Key Laboratory of Molecular and Cellular Biology, College of Life Sciences, Hebei Normal University, Shijiazhuang 050024, China

**Keywords:** extracellular ATP (eATP), *Arabidopsis thaliana*, AtANN3, auxin, seedling growth

## Abstract

Extracellular ATP (eATP) plays multiple roles in plant growth and development, and stress responses. It has been revealed that eATP suppresses growth and alters the growth orientation of the root and hypocotyl of *Arabidopsis thaliana* by affecting auxin transport and localization in these organs. However, the mechanism of the eATP-stimulated auxin distribution remains elusive. Annexins are involved in multiple aspects of plant cellular metabolism, while their role in response to apoplastic signals remains unclear. Here, by using the loss-of-function mutations, we investigated the role of AtANN3 in the eATP-regulated root and hypocotyl growth. Firstly, the inhibitory effects of eATP on root and hypocotyl elongation were weakened or impaired in the *AtANN3* null mutants (*atann3–1* and *atann3–2*). Meanwhile, the distribution of DR5-GUS and DR5-GFP indicated that the eATP-induced asymmetric distribution of auxin in the root tips or hypocotyl cells occurred in wild-type control plants, while in *atann3–1* mutant seedlings, it was not observed. Further, the eATP-induced asymmetric distribution of PIN2-GFP in root-tip cells or that of PIN3-GFP in hypocotyl cells was reduced in *atann3–1* seedlings. Finally, the eATP-induced asymmetric distribution of cytoplasmic vesicles in root-tip cells was impaired in *atann3–1* seedlings. Based on these results, we suggest that AtANN3 may be involved in eATP-regulated seedling growth by regulating the distribution of auxin and auxin transporters in vegetative organs.

## 1. Introduction

Primary messengers that exist in the apoplastic matrix are responsible for modulating cell metabolism, making the apoplast, which comprises the cell wall and intercellular spaces, an essential modulator of plant cell growth and development. Among these signaling molecules, eATP plays an essential role. Intracellular ATP can leak through plasma membrane (PM) wounds, be secreted in secretory vesicles, or be released through specialized PM transporters [1,2,3,4]. eATP is involved in maintaining cell viability and regulating the growth rate and direction of some vegetative organs such as roots and hypocotyls [5,6,7,8,9], as well as the reproductive processes [10,11]. eATP is also involved in regulating stomatal movement and gravitropism [12,13,14,15]. In response to (a)biotic stresses (e.g., cold, salt, or pathogen attack), ATP secretion can be increased, inducing the eATP-stimulated defense or tolerance responses that act as the “danger signal” [1,16,17,18,19,20,21,22]. As the main eATP-hydrolyzing enzyme, extracellular apyrase is involved in terminating the signal transduction pathway and maintaining the eATP levels [23,24].

To elucidate the mechanisms underlying the functions of eATP, signal transduction of eATP has been extensively investigated. The first step involved in the eATP signaling pathway is the binding of eATP to its receptors. Two lectin receptor-like kinases, P2K1 and P2K2, were identified in *Arabidopsis thaliana* as eATP receptors [21,25]. The two P2K receptors have been shown to collaboratively or independently participate in plant immune responses. Several signaling proteins in PM, e.g., heterotrimeric G proteins [7,14,26], NADPH oxidases [12,14,27], and ion channels [27,28,29,30,31], have been reported to be involved in eATP-stimulated physiological responses. These signal transducers are speculated to be involved in the eATP-stimulated generation of secondary messengers (e.g., Ca^2+^, nitric oxide, or reactive oxygen species) or intracellular signal transduction cascades [7,9,14,27,32,33,34]. eATP-induced gene expression was proposed to regulate plant growth and development in response to environmental stimuli [8,25,35].

Annexins are Ca^2+^/phospholipid-binding proteins that are located in PM or the inner membrane, as well as in the cytoplasm [36,37,38]. Normally, there are multiple members in the annexin family in each plant species, with different members having different subcellular localizations and physiological functions. Data obtained from plants that have been investigated so far showed that annexin family members are involved in seed germination and early seedling growth [39,40,41], the transition from vegetative to the reproductive phase [42], pollen germination and tube growth [43], etc. Biotic stresses (fungal or viral pathogen attack) and abiotic stresses (cold, heat, salt, and drought) promoted the expression of some members of the annexin gene family, and the expression levels of most annexin members were positively correlated with the tolerance of plant cells to, or their defense against, multiple environmental stresses [39,40,44,45,46,47,48,49].

Research results have so far directly or indirectly provided clues to the physiological functions of annexins. It has been reported that annexins are implicated in Ca^2+^ signaling, catalyzing metabolic reactions, and vesicle trafficking. Some annexin family members in maize and Arabidopsis have been shown to establish reactive oxygen species (ROS)-responsive Ca^2+^ or K^+^ channels [50,51,52,53]. Importantly, AtANN1 which acts as a ROS-stimulated Ca^2+^ channel was shown to play a major role in the eATP-induced increase in [Ca^2+^]_cyt_ in Arabidopsis roots [51,54,55,56]. AtAnn2/AtANN3-mediated Ca^2+^ signaling is involved in photoperiod regulated drought stress responses [57]. Certain annexin members with peroxidase activity are shown to be involved in cellular redox reactions, e.g., when plants are exposed to stresses, annexins may suppress ROS accumulation, reduce lipid peroxidation, and regulate activities of the cell [58,59]. Some annexin members are membrane lipid or cytoskeleton-binding proteins which localize in PM or the inner membrane and participate in cytoplasmic vesicle trafficking and cell secretion [41,43,60,61].

As a multifunctional plant hormone, auxin plays essential roles in regulating plant growth and development. Plants responding to external stimuli, such as light, gravity, and water, exhibited altered growth rates and orientation. Auxin accumulation and asymmetric distribution are responsible for regulating the elongation rate of plant cells in different parts of plant organs, which results in the bending of these organs [62,63]. Auxin transporters, especially PIN-FORMED transporters (PINs), play key roles in polar auxin transport [63,64,65]. The asymmetric distribution of PINs results in unidirectional auxin transport. After stimulation, subcellular PIN trafficking and PIN phosphorylation alter the localization of PINs, which would subsequently alter auxin transport [66,67,68,69,70]. The small G protein- or clathrin-mediated vesicle trafficking is involved in PIN trafficking during phototropic and gravitropic bending [71,72,73,74]. Most recently, two SNARE proteins were reported to be involved in auxin-regulated seedling growth via regulating subcellular trafficking of auxin transporters [70]. In response to endogenous or exogenous stimuli, several amino acids in the long central hydrophilic ring of PINs can be phosphorylated, and phosphorylation is sufficient to modulate the polar distribution, recycling, and turnover of PIN proteins in a ubiquitin-dependent manner [68,69,75]. There are eight members in the PIN family in *Arabidopsis thaliana*, with each having distinct spatial–temporal expression and location profiles. In *Arabidopsis thaliana* seedlings, PIN-mediated auxin redistribution plays essential roles in tropic responses of roots and hypocotyls to various stimuli [62,63,64,66].

eATP regulates the asymmetric distribution of auxin in roots of *Arabidopsis thaliana*, which alters the growth rate and direction of roots or hypocotyls. PIN2 and PIN3 have been reported to be involved in the eATP-regulated auxin transport [7,8,76]. However, the mechanism underlying the eATP-regulated PIN abundance and relocation, which in turn alters the auxin asymmetric distribution, remains unclear. Herein, to elucidate the role of annexins in eATP signaling, we investigated the effects of ATP supplementation on seedling growth and auxin distribution in *AtANN3* null mutant lines.

## 2. Results

### 2.1. AtANN3 Is Involved in Responses of Arabidopsis Seedlings to eATP Supplementation

In our previous work, we showed that *Arabidopsis thaliana* seedlings responded to eATP by altering the growth rate and orientation of their roots and etiolated hypocotyls [7,8]. To verify the role of AtANN3 in eATP signaling, responses of the seedlings of two *AtANN3* null mutants to ATP were investigated.

When seedlings of wild-type (Col-0) plants were transplanted onto the combined medium containing 0.5 mM ATP in the lower compartment, main roots showed a marked eATP avoidance response characterized by suppressed growth and altered growth direction as they approached the border between the two growth media (Figure 1A). The ATP-suppressed root elongation was also observed in seedlings of the two mutant lines, and their roots were also significantly shorter than those of control mutant seedlings. Seedlings of wild-type and both mutant lines had, statistically significantly, the same control root lengths. (Figure 1B). However, the eATP avoidance response of the seedlings of *atann3–1* and *atann3–2* mutants was significantly reduced compared to that of wild-type plants; the degree of the root curvature in the seedlings of the two mutants was markedly smaller than that of the wild-type Col-0 plants (Figure 1C).

To further verify the role of AtANN3 in responses to eATP addition, we examined the hypocotyl growth rate and curvature of etiolated seedlings, which were grown on the growth medium containing 0.5 mM eATP. Results showed that the seedlings of *atann3–1* and *atann3–2* responded to eATP by slightly suppressing the growth and small bending curvature of hypocotyls. Compared to Col-0, the growth of hypocotyls of the etiolated *atann3–1* and *atann3–2* seedlings was partially resistant to the eATP treatment; the hypocotyl was significantly longer, and the degree of hypocotyl curvature was significantly smaller than that in Col-0 (Figure 1D–F).

### 2.2. AtANN3 Is Involved in the eATP-Regulated Auxin Distribution

To clarify the role of AtANN3 in eATP-regulated seedling growth, the seedlings of *DR5-GUS* and *DR5-GFP* transgenic *atann3–1* plants were transplanted onto the medium containing ATP, and then GUS staining and confocal laser scanning microscopy (CLSM) were used to investigate the effects of eATP on the abundance and distribution of GUS or GFP in roots and etiolated hypocotyls.

GUS staining in Col-0 seedlings grown under light conditions showed that GUS expression was found mainly in root-tip cells, especially in cells around the quiescent center (QC). ATP treatment promoted GUS accumulation and caused the distribution of GUS from the root tips to meristematic and elongation zones, especially in the stele cells. In untreated *atann3–1* seedlings, GUS was expressed in root-tip cells and some stele cells, with an abundance similar to that in wild-type plants. After ATP treatment, GUS was accumulated in root-tip cells, but its abundance and the extent of its distribution were both lower than in wild-type plants, demonstrating the weakened response of *atann3–1* seedlings to eATP (Figure 2A). Intensity and localization of DR5-GFP in the root-tip cells were also markedly affected by the ATP addition. In ATP-treated Col-0 seedlings, the fluorescence intensity increased in QC, stele, and epidermal cells, and a marked asymmetric distribution in epidermal cells was observed (Figure 2B). At the root curvature, the fluorescence intensity in cells at the inner side was significantly higher than that in cells at the outer side (*p* < 0.05) (Figure 2C). In seedlings of ATP-treated *atann3–1* plants, the DR5-GFP fluorescence intensity slightly increased in cells around the QC, while the asymmetric distribution of DR5-GFP fluorescence in epidermal cells did not occur (Figure 2B,C).

In the hypocotyl of the etiolated Col-0 seedlings, GUS activity increased after ATP treatment, and a marked asymmetric distribution occurred at the bending area, with a higher GUS abundance in cells at the outer side than at the inner side. In ATP-treated *atann3–1* seedlings, the GUS accumulation or the asymmetric distribution did not occur (Figure 3A). The detection of DR5-GFP fluorescence further confirmed the effects of ATP. In ATP-treated Col-0 seedlings, DR5-GFP fluorescence was accumulated in hypocotyl cells, and an asymmetric fluorescence distribution was also detected (Figure 3B). At the bending area, the fluorescence intensity in the cells at the outer side was significantly higher than at the inner side (*p* < 0.01) (Figure 3C). In eATP-treated *atann3–1* seedlings, neither an intensity increased nor asymmetric distribution of fluorescence was detected (Figure 3B,C).

### 2.3. AtANN3 Is Involved in the eATP-Induced Auxin Transport and Distribution

To verify the role of AtANN3 in the eATP-regulated auxin transport, *PIN2-GFP* and *PIN3-GFP* transgenic wild-type and *atann3–1* plants were used to investigate the effects of eATP on the abundance and distribution of the two auxin transporters in seedlings.

In seedlings grown under light conditions, after ATP stimulation, a marked asymmetric distribution of PIN2-GFP in root-tip cells of Col-0 plants occurred (Figure 4A). At the bending area, the fluorescence intensity in epidermal cells at the inner side was significantly greater than at the outer side. Data analysis showed that the fluorescence intensity ratio (FIR) at the inner side/outer side significantly increased after ATP stimulation (*p* < 0.05) (Figure 4C). By contrast, in *atann3–1* roots, the asymmetric distribution of PIN2-GFP was not observed (Figure 4A,C).

In untreated seedlings, PIN3-GFP fluorescence was mainly observed in the QC and stele cells. After ATP stimulation, the fluorescence intensity of PIN3-GFP in Col-0 root-tip cells markedly decreased in both the QC and stele cells (Figure 4B). In *atann3–1* seedlings, ATP treatment led to a significant decrease in PIN3-GFP fluorescence intensity. The degree of reduction in PIN3-GFP fluorescence intensity was not significantly different between Col-0 and *atann3–1* seedlings (Figure 4D).

In hypocotyl of etiolated seedlings, the fluorescence intensity was very weak (a thin layer of fluorescence appeared on the surface of hypocotyl, which was unlikely to belong to any cell structure). After ATP stimulation, the fluorescence intensity and distribution did not change, in both the Col-0 and *atann3–1* seedlings (Figure 5A,C). PIN3-GFP fluorescence in etiolated hypocotyls of Col-0 was accumulated in the epidermal, cortex, and stele cells. After ATP treatment, the fluorescence intensity slightly decreased in epidermal and cortex cells, but it did not change in stele cells. At the bending area of curved hypocotyls, the fluorescence intensity in the epidermal and cortex cells at the outer side was significantly stronger than at the inner side. Data analysis showed that the fluorescence intensity ratio at the outer side/inner side significantly increased after ATP stimulation (*p* < 0.05). By contrast, in hypocotyls of the etiolated *atann3–1* seedlings, the PIN3-GFP fluorescence intensity did not change after ATP treatment and asymmetric distribution of PIN3-GFP also did not occur; moreover, the fluorescence intensity ratio did not significantly change after ATP stimulation (*p* > 0.05) (Figure 5B,D).

### 2.4. AtANN3 Is Involved in the eATP-Induced Vesicle Abundance in Root Cells

To verify the role of AtANN3 in the eATP-regulated distribution of auxin transporters, seedlings of *PIN2-GFP* transgenic lines, which were grown on combined medium containing 0.5 mM ATP were stained by FM 4–64, and green and red fluorescence emissions were detected simultaneously to verify the abundance of PIN2-GFP-containing vesicles. The results showed that in untreated Col-0 seedlings, the abundance of vesicles in cells at the inner side and outer side was similar. ATP stimulation led to a decrease in PIN2-GFP-containing vesicle abundance (Figure 6A), and the vesicle abundance in cells at the outer side was significantly lower than at the inner side (Figure 6B). In *atann3–1* seedlings, the PIN2-GFP-containing vesicle abundance was lower than in Col-0 seedlings, while there was no significant difference between the values in cells at the outer and inner side. After ATP treatment, the abundance of vesicles significantly decreased in cells at both the outer and inner side. Treatment with Brefeldin A (BFA) did not significantly affect vesicle abundance in general, either in control Col-0 or in control atann3-1 seedlings. However, BFA treatment blocked the eATP-induced asymmetrical localization of PIN2-GFP-containing vesicles in wild-type root tips and blocked the eATP-induced reduction in the abundance of PIN2-GFP-containing vesicles in annat3-1 root tips (Figure 6A,B).

## 3. Discussion

For eATP-regulated plant growth and development, the growth of root and hypocotyl was intensively investigated. High concentrations of eATP suppressed root elongation, leading to root bending and skewing or curling growth [5,6,15,26,76]. In our previous work, the main root of *Arabidopsis thaliana* responded to millimolar ATP stimulation by decreasing growth rate and markedly bending, which is termed as “eATP avoidance”, confirming that plant cells perceive high concentrations of eATP as a “danger signal” [7,8]. The eATP addition also effectively regulated hypocotyl growth and altered the growth rate and direction of etiolated hypocotyls, although the physiological relevance of the eATP-stimulated bending of hypocotyls remains unclear [8,9].

The responses of roots and hypocotyls to eATP simulation can be used as readouts for identifying the components involved in eATP signal transduction. In the untreated growth medium, the growth rate and direction of seedlings of *AtANN3* mutants were similar to those of wild-type controls, indicating that the loss-of-function mutants of *AtANN3* did not significantly affect seedling growth. After ATP addition, *atann3–1* and *atann3–2* seedlings exhibited a weaker response to eATP in terms of root and hypocotyl growth, suggesting that AtANN3 is involved in the eATP-regulated seedling growth. The role of AtANN3 in plant growth and development is unknown, although it is implicated in the formation of vacuoles by the fusion of vesicles in root cells [61]. Here, we present novel evidence for the role of AtANN3 in regulating seedling growth.

eATP stimulation leads to asymmetric auxin transport in root cells. Here, we further showed that the asymmetric distribution of auxin also occurred in the eATP-treated etiolated hypocotyls, which led to the bending of hypocotyls. Since root cells are very sensitive to auxin, the accumulated auxin in root cells would suppress the elongation of epidermal cells. By contrast, in hypocotyl cells, which are less sensitive to auxin than root cells, the accumulation of auxin would promote cell elongation. It has been reported that eATP stimulated the asymmetric auxin distribution in Arabidopsis seedlings [6,76]. Our finding indicating that AtANN3 is involved in the eATP-regulated auxin distribution is useful for understanding the mechanism of the eATP-stimulated auxin redistribution in vegetative organs.

The expression of some annexin genes has been revealed to be regulated by plant hormones, e.g., ABA, ethylene, or auxin, suggesting that annexins may be involved in plant-hormone-regulated growth, development, and stress responses [36,49,54,77]. Some annexins have been revealed to be involved in the regulation of auxin transport and distribution. For instance, AtANN1 and AtANN2 are involved in unilateral blue-light-induced asymmetric distribution of auxin and PINs in etiolated hypocotyls [78] and MeANN2 in cassava, which is similar to AtANN1, is involved in stress tolerance via regulating auxin signaling [79]. Our results provide direct evidence for the involvement of AtANN3 in regulating auxin transport in response to an extracellular signal.

Auxin efflux transporters, especially PINs, play essential roles in polar auxin transport and physiological responses induced by asymmetric distribution. In response to stimuli, the abundance and distribution of PINs dynamically change. Two types of vesicle transport, including endocytosis and exocytosis, are involved in protein relocation from one part of the cell to another. In *Arabidopsis thaliana*, PIN2 and PIN3 are involved in tropic responses of root and hypocotyl, respectively. During root phototropism or gravitropism, PIN2 mediates the asymmetric distribution of auxin in root-tip cells, especially in the elongation zone [69,80]. PIN3 is the main mediator of the lateral flow of auxin during hypocotyl phototropism or gravitropism [63]. Unidirectional blue light stimulated the movement of PIN3 from the irradiated side to the shaded side of etiolated hypocotyl cells [71]. During hypocotyl gravitropism, PIN3-mediated auxin directional flux and accumulation at the bottom side of the cells resulted in negative gravitropic bending [72]. Consistent with these reports, our previous work revealed that PIN2 is involved in eATP-induced root bending, while PIN3 is involved in eATP-induced hypocotyl bending [8]. The results of this study further confirm the role of the two PINs in the eATP-regulated growth of vegetative organs.

Subcellular trafficking is involved in PIN2 redistribution and subsequent auxin asymmetric distribution which lead to the growth of roots away from light or salinity [81,82,83,84,85]. PIN3 redistribution also results from subcellular trafficking [72,73,86]. AtANN3 is involved in regulating the vesicle transport from the Golgi apparatus to vacuoles [61]. The ATP-stimulated asymmetric distribution of auxin and auxin transporters in root-tip cells exposed to eATP was not observed in the *atann3–1* mutant, suggesting that the AtANN3-regulated PIN transport and asymmetric auxin distribution might be the important events in eATP signal transduction. In this work, eATP stimulation led to a decrease in the PIN2-GFP fluorescence intensity and an asymmetric distribution of the PIN2-GFP-containing vesicles that are abundant in root tips of wild-type plants but not in *atann3–1* seedlings. BFA treatment blocked the eATP-induced asymmetrical localization of PIN2-GFP-containing vesicles in wild-type root tips and blocked the eATP-induced reduction in the abundance of PIN2-GFP-containing vesicles in *annat3-1* root tips, indicating that AtANN3-involved vesicle trafficking is BFA-sensitive. BFA-sensitive and BFA-insensitive vesicle trafficking have been revealed to be involved in transport and recycling of auxin transporters (e.g., PINs and ABCBs) [87,88]. Hence, the results provided valuable clues about the role and possible mechanism of AtANN3 in the trafficking of vesicles which would lead to the recycling of auxin transporters, although the relationship between the localization of PIN proteins and the abundance and distribution of vesicles undergoing trafficking needs to be further clarified.

## 4. Materials and Methods

### 4.1. Plant Materials

*Arabidopsis thaliana* L. wild-type (Col-0) and two lost-of-function mutants of *AtANN3*, *atann3–1* (salk_082344) and *atann3–2* (salk_130101C), were used in this study. Seeds were obtained from ABRC (Arabidopsis Biological Resource Center, the Ohio State University, USA), and plants were genotyped to confirm the homozygous mutation of the gene.

### 4.2. Root or Hypocotyl Growth Determination

Seeds were surface-sterilized with 70% ethanol for 2 min and then with 5% sodium hypochlorite for 5 min. After two times washing with sterilized water, seeds were sown on the surface of the solid medium containing 1/2 MS (Murashige and Skoog) salt (Sigma-Aldrich), and 0.8% phytagel in square culture dishes. The culture dishes were stored at 4 °C for 2 days, and then seedlings were vertically cultured at 22 °C with a light intensity of 130 µmol·m^−2^·s^−1^ in a 16/8 h light/dark cycle.

To investigate the responses of roots to eATP addition, seedlings were grown on a combined medium made according to the protocol described by Zhu et al. [7,8]. A solution of 1/2 MS salt and 0.8% phytagel was sterilized and poured into 10 × 10 cm square culture dishes, with each dish containing 50 mL liquid medium. After solidifying the medium, it was cut along the midline of the culture dish with a sterilized blade, and half of the medium was removed. The interspace was then refilled with 25 mL of the sterilized 1/2 MS medium containing 0.5 mM ATP. ATP was dissolved in 1/2 MS solution to make a stock solution of 50 mM, and the pH value was adjusted to 6.0 with Tris. The stock solution was then filtered with a sterilized filter (SLGP033RB, 0.22 µm, Millipore, Burlington, MA, USA) and mixed with the sterilized 1/2 MS medium, which was cooled to 50 °C and poured into the interspace in the culture dish. After solidification, the refilled medium level was similar to that of the original medium. Seedlings grown on the 1/2 MS medium for 4 days were transplanted onto the untreated part of the combined medium, with the root tip toward the refilled medium and 0.3–0.5 cm from the border between the two growth media. The culture dishes were placed vertically, with the untreated part placed on top and the refilled medium at the bottom so that the root would grow downward toward the refilled part where ATP was present in the medium.

To determine the hypocotyl growth in etiolated seedlings, surface-sterilized seeds were sown on the surface of the solid 1/2 MS medium containing 0.8% phytagel supplemented with ATP in square culture dishes. The culture dishes were stored at 4 °C for 2 days, and then the seedlings were vertically cultured at 22 °C with a light intensity of 130 µmol· m^−2^. s^−1^ in a 16/8 h light/dark cycle for 24 h. Thereafter, culture dishes were coated with tin foil and placed vertically for the further culture of seedlings at 22 °C.

To measure the root or hypocotyl length and curvature, photos of seedlings were captured using an optical scanner and then analyzed using ImageJ software. The angle between the bending roots and the vertical direction was considered and measured as the curvature. In each experiment, at least 30 seedlings were measured, and the means of treatment groups were calculated from three replicates. Data were statistically analyzed using Sigmaplot version 12.0 and SPSS software (version 19.0).

### 4.3. GUS Histochemical Staining

The published *DR5-GUS* transgenic lines [8] were hybridized with *atann3–1*, and the expression of genes was detected by PCR. Seedlings were grown on the 1/2 MS medium containing ATP and cultured under light conditions for root staining and in darkness for hypocotyl staining. Seedlings were then collected and incubated at 37 °C in the GUS staining solution containing 1 mM X-Gluc, 50 mM PBS, 10 mM EDTA, 0.1% Triton X-100, and 0.5 mM potassium ferricyanide. After a certain period, the stained seedlings were transferred into absolute ethanol to decolorize them until the tissues became transparent. Images were captured using a stereomicroscope (Olympus SZX16, Tokyo, Japan).

### 4.4. Confocal Laser Scanning Microscopy (CLSM)

Published *DR5-GFP*, *PIN2-GFP*, and *PIN3-GFP* transgenic lines [89] (seeds were provided by Dr. Jiří Friml in the Institute of Science and Technology Austria, Klosterneuburg, Austria) were hybridized with *atann3–1*, and the expression of genes was detected by PCR. The 4-day-old seedlings were transplanted onto the medium containing ATP and cultured for 2 more days. Seedlings with bending roots or hypocotyls from both wildtype and mutants were collected (mutant seedlings, while showing some bending, did not show as much bending as wild-type seedlings). Seedlings were placed onto the microscope stage equipped with a laser confocal scanning system (LSM 710, Zeiss, Jena, Germany), and fluorescence in cells at the bending area was photographed. The excitation and emission wavelengths were 488 nm and 515 nm, respectively. Images were processed with Confocal Assistant 4.0 software and edited with Adobe Photoshop (version 2021).

To measure the fluorescence intensity, a region of interest 0.5 mm long in root-tip cells was delineated. To obtain the fluorescence intensity ratio, cells on the left side and right side of the root or hypocotyl were delineated as the area of interest. Fluorescence intensity was measured with ImageJ software, and then the data were statistically analyzed using Sigmaplot (version 12.0) and SPSS (version 19.0) software programs.

### 4.5. Cytoplasmic Vesicle Amount Measurement

The 4-day-old *PIN2-GFP* transparent seedlings of Col-0 or *atann3-1* were transplanted onto combined medium and cultured vertically under light for 24 h. To verify the effect of vesicle trafficking inhibitor, 50 μM Brefeldin A (Sigma) was added to the medium with or without ATP. Seedlings with bending roots from both wildtype and *atann3-1* were selected and stained with 20 μg·mL^−1^ FM 4–64 to visualize the plasma membrane and intracellular vesicles. Root tips were placed onto the microscope stage equipped with a laser confocal scanning system (LSM 710, Zeiss, Jena, Germany), and fluorescence in cells at the inner side (IS) and the outer side (OS) of the bending area was captured. The number of vesicles was counted manually and then the data were statistically analyzed by using Sigmaplot (version 12.0) and SPSS (version 19.0) software programs.

## Figures and Tables

**Figure 1 plants-12-00330-f001:**
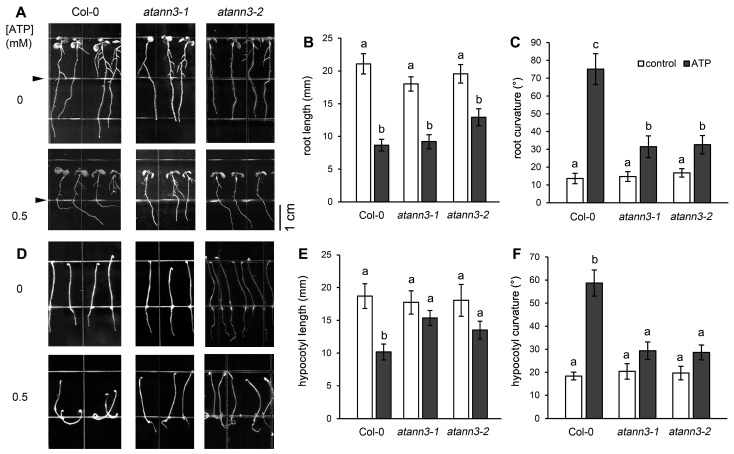
AtANN3 is involved in the eATP-regulated seedling growth of *Arabidopsis thaliana*. (**A**–**C**) The 4-day-old seedlings were transplanted onto the combined medium and cultured under light conditions for 5 more days. The concentration of ATP in the lower compartment was 0.5 mM. The triangles mark the border between the two growth media. (**A**) The growing seedlings and (**B**,**C**) the root length and root curvature of seedlings, respectively. (**D**–**F**) Seeds were sown and cultured on the medium containing 0.5 mM ATP in the dark for 5 days. (**D**) The growing etiolated seedlings and (**E**,**F**) the hypocotyl length and curvature of seedlings, respectively. The scale bar is shown in the lower-right corner of (**A**). In each experiment, at least 30 seedlings were measured, and data obtained from at least three replicates were combined to obtain mean ± SD and then analyzed statistically. The same letter on the top of the columns indicates that means of different treatment groups were not significantly different based on Duncan’s multiple range test (*p* < 0.05).

**Figure 2 plants-12-00330-f002:**
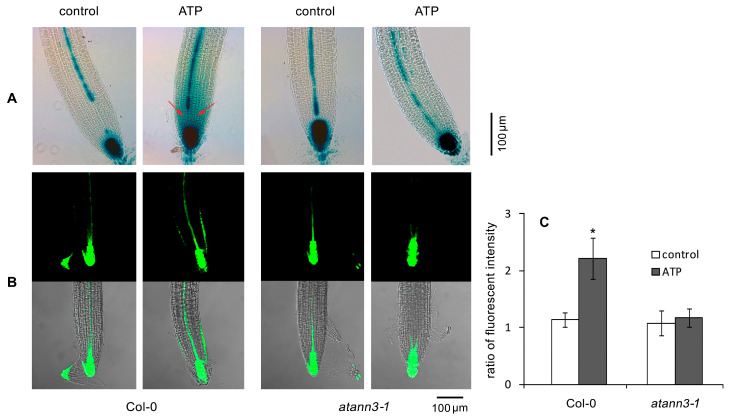
AtANN3 is involved in the eATP-induced distribution of DR5-GUS or DR5-GFP in root-tip cells. The 4-day-old seedlings of *DR5-GUS* or *DR5-GFP* transgenic lines were transplanted onto the medium containing 0.5 mM ATP and cultured for 2 more days. (**A**) The *DR5-GUS* expression in representative root-tip cells; red arrows indicate the area with the accumulation of DR5-GUS. (**B**) The DR5-GFP fluorescence in representative root-tip cells. The scale bar is shown in the lower-right corner of the figure. (**C**) The fluorescence intensity ratio (FIR) in the cells at the inner side/outer side of the root curve. In this experiment, at least 15 samples were measured, and data obtained from at least three replicates were combined to obtain mean ± SD based on Student’s *t*-test (* *p* < 0.05).

**Figure 3 plants-12-00330-f003:**
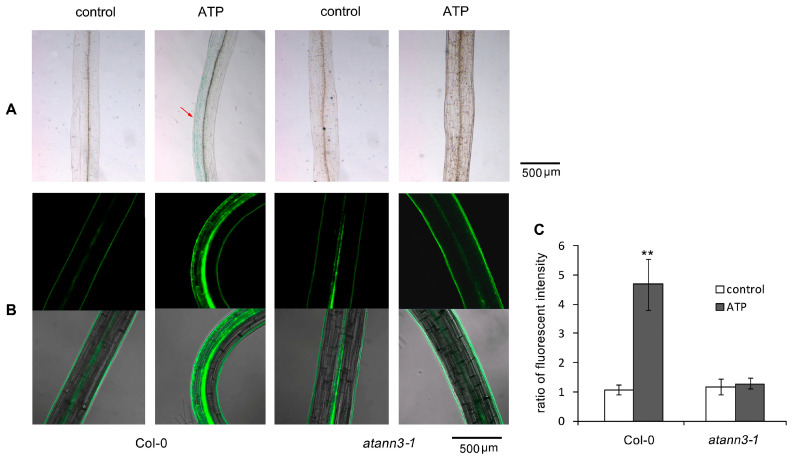
AtANN3 is involved in the eATP-induced DR5-GUS or DR5-GFP distribution in the etiolated hypocotyl cells. The 4-day-old etiolated seedlings of *DR5-GUS* or *DR5-GFP* transgenic lines were transplanted onto the medium containing 0.5 mM ATP and cultured for 2 more days. (**A**) The *DR5-GUS* expression in representative hypocotyls. The red arrow indicates the accumulation area of DR5-GUS. (**B**) The DR5-GFP fluorescence in representative hypocotyls. The scale bar is shown in the lower-right corner of each figure. (**C**) The fluorescence intensity ratio (FIR) in cells at the inner side/outer side at the hypocotyl curve. In this experiment, at least 15 samples were measured, and data from at least three replicates were combined to obtain mean ±SD based on Student’s *t*-test *p*-values (** *p* < 0.01).

**Figure 4 plants-12-00330-f004:**
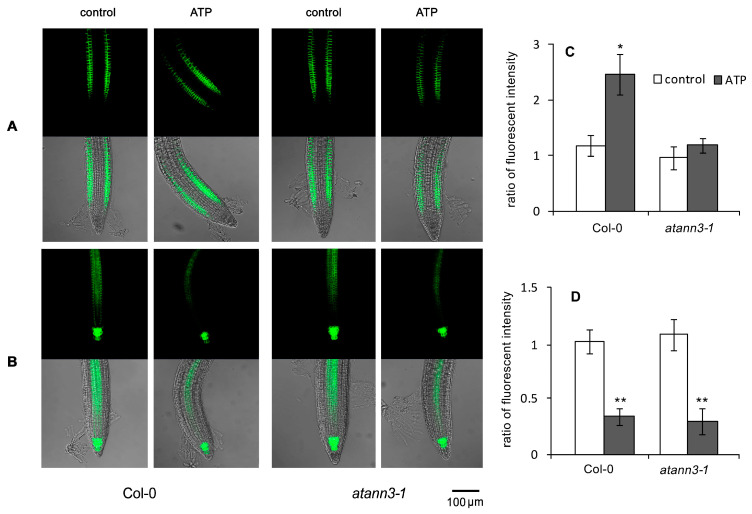
AtANN3 is involved in the eATP-induced PIN2-GFP distribution in root cells. The 4-day-old seedlings of *PIN2-GFP* or *PIN3-GFP* transgenic lines were transplanted onto the medium containing 0.5 mM ATP and cultured for 2 more days. (**A**,**B**) Fluorescent images of PIN2-GFP and PIN3-GFP in root tips, respectively. The scale bar is shown in the lower-right corner of the image. (**C**) The fluorescence intensity ratio of PIN2-GFP in cells at the inner side/outer side at the root curve. (**D**) The fluorescence intensity ratio of PIN3-GFP in QC and stele cells before and after the ATP treatment. The Fluorescence intensity in root tips of untreated Col-0 seedlings was set at 1. In each experiment, at least 15 samples were measured and data from at least three replicates were combined to obtain mean ± SD based on Student’s *t*-test *p*-values (* *p* < 0.05, ** *p* < 0.01).

**Figure 5 plants-12-00330-f005:**
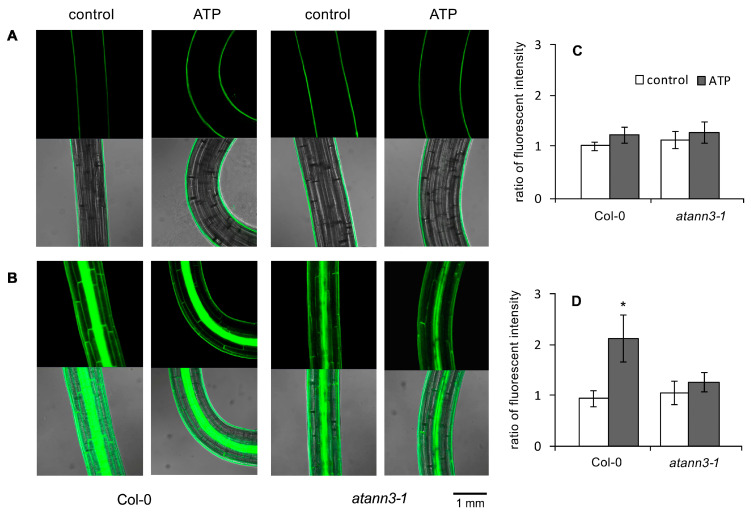
AtANN3 is involved in the eATP-induced PIN3-GFP distribution in hypocotyl cells. The 4-day-old seedlings of *PIN2-GFP* or *PIN3-GFP* transgenic lines were transplanted onto the medium containing 0.5 mM ATP and cultured for 2 more days. (**A**,**B**) Fluorescence in the hypocotyl of PIN2-GFP (**A**) and PIN3-GFP (**B**) transformant seedlings. The scale bar is shown in the lower-right corner of the image. (**C**,**D**) The fluorescence intensity ratio in cells at the inner side/outer side at the hypocotyl curve of PIN2-GFP (**C**) and PIN3-GFP (**D**) transformant seedlings. In each experiment, at least 15 samples were measured, and data from at least three replicates were combined to obtain mean ± SD based on Student’s *t*-test *p*-values (* *p* < 0.05).

**Figure 6 plants-12-00330-f006:**
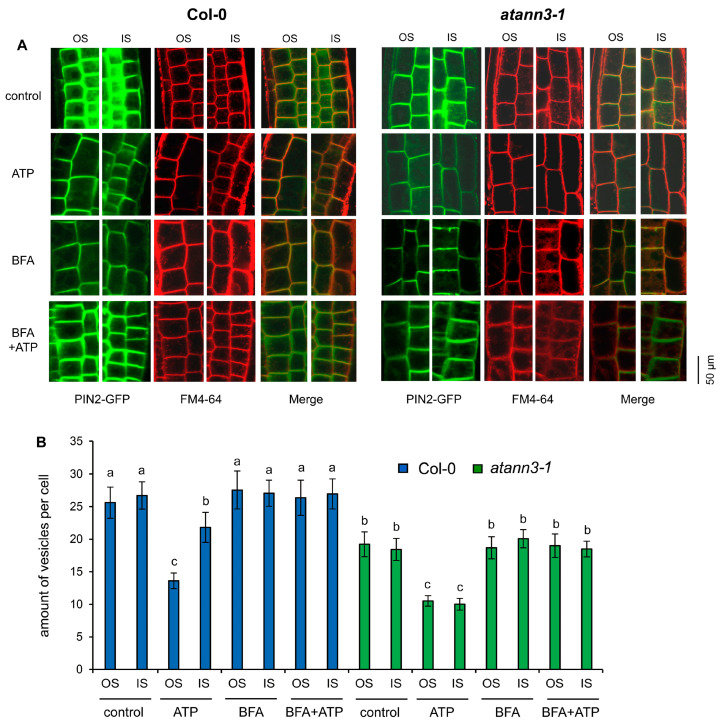
AtANN3 is involved in the eATP-regulated PIN2-GFP-containing vesicle abundance in root-tip cells. (**A**) Images of the cells at the inner side (IS) and the outer side (OS) of the bending area and (**B**) the number of vesicles in root cells. In each experiment, at least 40 cells in 5–6 seedlings were measured, and data from three replicates were combined to obtain mean ±SD and statistically analyzed. The same letter on the top of the columns indicates that the means of different groups were not significantly different based on Duncan’s multiple range test (*p* < 0.05).

## Data Availability

Not applicable.

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
