# Peer review of "ATANN3 Is Involved in Extracellular ATP-Regulated Auxin Distribution in Arabidopsis thaliana Seedlings"

_plants, 2023, doi:10.3390/plants12020330_

Round 1

Reviewer 1 Report

This manuscript submitted by Xu et.al describes differences in the response of Arabidopsis wild-type and annexin 3 loss-of-fun function mutant seedlings to ATP treatments. Dark-grown wild-type hypocotyls show a statistically significantly ATP-induced inhibited growth and increased hypocotyl curvature while annat3 knockout seedlings do not. Light-grown wild-type and annat3 knockout roots both show a statistically significantly ATP-induced inhibited growth. The ATP-induced root avoidance response is reduced in annat3 knockout seedlings compared to wild-type. This study also examined differences in the distribution of auxin reporters, PIN auxin transporters, and FM-64 vesicle staining in wild-type and mutant roots and hypocotyls. 

 Major issues:

1) Lines 102-103 states “Root length of the 2 mutant lines 102 were significantly shorter than that of control (Fig. 1B).” This is an incorrect statement based on the data shown in the Figure. The average root length of both wild-type and annat3 seedlings are both statistically significantly shorter in response to ATP treatment and based on the letters of significance are the statistically the same in controls and ATP conditions.

2) Although Fig. 1 shows quantification of differences in root and hypocotyl curvature between wild-type and annat3 seedlings there seems to be examples of mutant seedlings which show tissue curvature responses. In the experiments performed in Figs. 4-6, the mutant roots and hypocotyls chosen seem to be from the non-curving seedlings and instead they should be seedlings that are showing some curvature. An image of the wild-type and mutant seedlings chosen for the microscopy experiments should be included in Figs. 4-6 with an arrow indicating where the corresponding microscopic images were taken from in the respective roots and hypocotyls.

3) Fig. 6 shows important data but is missing a key control of treatment with Brefeldin A (BFA) which is an inhibitor of vesicle trafficking. The authors need to do this control to confirm the results shown in Fig. 6.

 Minor issues:

1) The use of English is problematic throughout this manuscript so the text requires extensive editing by a native English speaker.

2) The authors need to add text to the Introduction and Discussion sections better describing and discussing previous studies that found an intersection between annexin and extracellular ATP signaling. They cite Laohavisit et al., 2014 but do not discuss the findings of this study. Additionally, they do not cite or discuss the findings from Mohammad-Sidik et al., 2021 (see reference below).  

Mohammad-Sidik, A.; Sun, J.; Shin, R.; Song, Z.; Ning, Y.; Matthus, E.; Wilkins, K.A.; Davies, J.M. Annexin 1 Is a Component of eATP-Induced Cytosolic Calcium Elevation in Arabidopsis thaliana Roots. Int. J. Mol. Sci. 202122, 494. https://doi.org/10.3390/ijms22020494

3) The authors state on page 10, line 254 “However, annexin-regulated transport of plant hormone has rarely been reported.”. Please correct this statement and add text describing the findings in Wang et al., 2022 (see reference below).

Xiaoxu Wang, Lijuan Han, Hongmin Yin, Zhenping Zhao, Huishu Cao, Zhonglin Shang, Erfang Kang, AtANN1 and AtANN2 are involved in phototropism of etiolated hypocotyls of Arabidopsis by regulating auxin distribution, AoB PLANTS, Volume 14, Issue 1, February 2022, plab075, https://doi.org/10.1093/aobpla/plab075

Reviewer 2 Report

In Jiawei Xu et al., authors suggest a role of the extracellular ATP response via annexins3 in governing plant growth. This mechanism is determined by a regulation of auxin distribution by controlling the polar auxin transporters PIN3 and PIN2. Here follow my concerns:

·       The introduction lacks information on the already known AtANN3 activity during developmental processes. Authors only refer to the entire family. I suggest providing insights into the AtANN3 function.

·       did authors verify that atann3-1 (salk_082344) and 282 atann3-2 (salk_130101C) were loss of function mutants?

·       Line 109: “were more resistant” I would say “partially resistant”

·       please indicate the statistical test used for the experiments reported in Fig.1 and Fig.6

·       DR5-GFP, DR5-GUS, PIN2-GFP, PIN3-GFP: plants were transformed with or crossed with? Please correct it

·       lines 129-131: please correct typos

·       lines 131-132 and Fig.2A: given the different focal plane between col and atann3-1 roots, authors cannot state that atann3-1 seedlings show a weakened response to eATP. Indeed, gus expression appears increased in the vasculature of atann3-1. Please comment on it.

·       Lines 133-135 and Fig.2B: knowing that ATP treatment determines an agravitropic response, this would cause the DR5-GFP asymmetric distribution observed in col. It is not clear what authors want to highlight with this sentence

·       Lines 133-135 and Fig.2B,C: as before, knowing that atann3-1 showed a weak response to ATP treatment, this would result in no alteration of the DR5-GFP distribution. It is not clear what authors want to highlight with this sentence, maybe to emphasize that ATP treatment would affect auxin distribution which in turns direct root gravitropic response, they could rephrase the sentence

·       Fig.3B,C: in the vasculature of ATP-treated atann3-1 the DR5-GFP expression seems not present. How could authors justify it?

·       it is not clear why authors focus on PIN3 expression in the atann3-1 root. Please clarify it

·       lines 172-173: it is really hard to have such conclusion without a measurement of PIN2 expression in treated and untreated plants, please provide a fluorescent quantification

·       fig.4A: same as before: knowing that ATP treatment determines an agravitropic response, this would cause the PIN2-GFP asymmetric distribution observed in col. It is not clear what authors want to highlight with this sentence

·       line 176: to substantiate the sentence, please provide a PIN2 fluorescent quantification of atann3-1 mutants treated and untreated

·       fig.4A,C: as before, knowing that atann3-1 showed a weak response to ATP treatment, this would result in no alteration of the PIN2-GFP distribution. It is not clear what authors want to highlight with this sentence, maybe to emphasize that ATP treatment would affect auxin distribution which in turns direct root gravitropic response, they could rephrase the sentence

·       fig.4D: it would be more informative to show the quantification of GFP signal in both line and not presenting only the ratio after/before in order to compare PIN expression in col and mutant

·       lines 192-194: it is not clear why authors focus on PIN2 expression in the atann3-1 hypocotyl. Please clarify it.

·       Lines 195-196: In figure 5D authors only showed a fluorescence signal ratio inner/outer sides. To substantiate this conclusion, I suggest to provide a separate fluorescence quantification of vasculature and epidermis

·       To support authors statements of PIN2 and PIN3 regulation via ANN3, understand where ANN3 localize in the root and in the hypocotyl would be crucial

·       Line 212: “as material”, please rephrase it

·       Lines 214-216: this is not clear to me. Do the authors have any data or/and reference for these claims? please clarify it

·       the experiment reported in the paragraph “2.4. AtANN3 is involved in eATP-induced vesicle abundance in root cells” is not described in a comprehensive way. It is unclear how the authors carried this out. Please report it in detail as no materials and methods section is provided about it. Moreover, the images reported in Fig6 are not understandable and explanatory.

·       Several papers already report and discuss the correlation between eATP and auxin (e.g 10.1104/pp.106.085670, 10.1186/1477-5956-9-72, 10.1083/jcb.142.6.1413, 10.1105/tpc.008433, 10.3390/ijms18040863), as also authors state in the text “It had been reported that eATP stimulates asymmetric auxin distribution in Arabidopsis seedlings”, it probably doesn't seem very clear what is the novelty of this manuscript. In my opinion, the authors could stress more the concepts they would like to emphasize.

Round 2

Reviewer 1 Report

The authors of this manuscript addressed some of my recommended changes and the manuscript has been improved. However, from my recommendations in my first review, there is still one experiment that is required before the manuscript can be accepted. There are also so more minor changes still needed to the text of the manuscript. Please see major and minor issues listed below.

Major issues:

Original comment:

1) Lines 102-103 states “Root length of the 2 mutant lines were significantly shorter than that of control (Fig. 1B).” This is an incorrect statement based on the data shown in the Figure. The average root length of both wild-type and annat3 seedlings are both statistically significantly shorter in response to ATP treatment and based on the letters of significance are the statistically the same in controls and ATP conditions.

Authors' Reply: The root elongation of wild type and the two mutants’ seedlings was inhibited in same degree. That mean there was no difference before and after ATP treatment. It does not mean the response of the two mutants to ATP was significantly different from wild type seedlings. We have revised the sentence as “The ATP-suppressed root elongation was also observed in seedlings of the 2 mutant lines, and their roots were also significantly shorter than those of control plants.” (Line 119-120.)

My reply:

1. Your revised text (lines 119-20) is an improvement but to make it even clearer I recommend changing the text further: “The ATP-suppressed root elongation was also observed in seedlings of the 2 mutant lines, and their roots were also significantly shorter than those of control mutant seedlings. Seedlings of wild-type and both mutant lines had statistically significantly the same control root lengths.”  

Original comment:

2) Although Fig. 1 shows quantification of differences in root and hypocotyl curvature between wild-type and annat3 seedlings there seems to be examples of mutant seedlings which show tissue curvature responses. In the experiments performed in Figs. 4-6, the mutant roots and hypocotyls chosen seem to be from the non-curving seedlings and instead they should be seedlings that are showing some curvature. An image of the wild-type and mutant seedlings chosen for the microscopy experiments should be included in Figs. 4-6 with an arrow indicating where the corresponding microscopic images were taken from in the respective roots and hypocotyls.

Authors' Reply: The images are do photos of bending area of the seedlings. Since the curvature of the mutants’ seedlings was markedly smaller than the wildtype, they looked less curved. Nevertheless, the bending of root tips or hypocotyls (to the right), especially of eATP-treated seedlings, were clearly showed in the figures. 

When we captured fluorescent images, only part of the seedlings can be photographed by using CLSM. The photo of whole plant was not captured. We apologize for not being able to provide these photos.

My reply:

2. Although it would have been much better to image each seedling on the plate before using it for microscopy. I think adding a description to the Material and Methods section to clarify to your audience the basis of how seedlings were chosen for microscopic analyses i.e that you chose bending seedlings from both wild-type and mutant lines but mutant seedlings while showing some bending did not show as much bending as wild-type seedlings 

Original comment:

3) Fig. 6 shows important data but is missing a key control of treatment with Brefeldin A (BFA) which is an inhibitor of vesicle trafficking. The authors need to do this control to confirm the results shown in Fig. 6.

Authors' Reply: Thanks for your advice. Yes, data in Fig.6 are still quite preliminary. We will conduct more test to verify the role of AtANN3 in vesicle trafficking in future work. Before we submit the manuscript, we once hesitated whether the figure should be included. Anyway, since the figure can provide some clues, putting it in the manuscript may be of some help.

My reply:

3. This experimental data needs to be included in the manuscript. However, in order to include it the authors will need to perform the key control of treating both wild-type and mutant controls and ATP-treated with and without Brefeldin A (BFA). 

Minor issues:

Original comment:

1) The use of English is problematic throughout this manuscript so the text requires extensive editing by a native English speaker.

Authors' Reply: The use of English has been completely revised by an English-speaking scholar from a professional editing group during the past week.

My reply:

1. Text has been greatly improved but there is still a need for more proofreading. For example, pg 11, line 280 change “plant growth & development” to “plant growth and development”. Another example, on pg 3, line 79 change “AtAnn2-/ AtANN3-mediated” to “AtANN2/AtANN3-mediated”.

Original comment:

2) The authors need to add text to the Introduction and Discussion sections better describing and discussing previous studies that found an intersection between annexin and extracellular ATP signaling. They cite Laohavisit et al., 2014 but do not discuss the findings of this study. Additionally, they do not cite or discuss the findings from Mohammad-Sidik et al., 2021 (see reference below).  

Mohammad-Sidik, A.; Sun, J.; Shin, R.; Song, Z.; Ning, Y.; Matthus, E.; Wilkins, K.A.; Davies, J.M. Annexin 1 Is a Component of eATP-Induced Cytosolic Calcium Elevation in Arabidopsis thaliana Roots. Int. J. Mol. Sci. 2021, 22, 494. https://doi.org/10.3390/ijms22020494

Authors' Reply: Thanks for your kind advice. We have added the discussion about the results in related publications (Mohammad-Sidik et al. 2021; Richards et al. 2014; Laohavisit et al.2009,2012) in the introduction. (Line 70-74)

My reply:

2. It was important to add this citation and reference but the new text on pg 2, lines 70-74 needs to be improved. I recommend changing “(ROS)-responsive Ca2+ or K+ channels [50–53], e.g., AtANN1 in Arabidopsis thaliana acting as a ROS-stimulated Ca2+ channel was revealed to be involved in eATP-induced Ca2+ signaling and probably also in stress-induced Ca2+ signaling [51, 54–56].” to “(ROS)-responsive Ca2+ or K+ channels [50–53]. Importantly, AtANN1 which acts as a ROS-stimulated Ca2+ channel was shown to play a major role in the eATP-induced increase in [Ca2+]cyt in Arabidopsis roots [55]. 

Original comment:

3) The authors state on page 10, line 254 “However, annexin-regulated transport of plant hormone has rarely been reported.”. Please correct this statement and add text describing the findings in Wang et al., 2022 (see reference below).

Xiaoxu Wang, Lijuan Han, Hongmin Yin, Zhenping Zhao, Huishu Cao, Zhonglin Shang, Erfang Kang, AtANN1 and AtANN2 are involved in phototropism of etiolated hypocotyls of Arabidopsis by regulating auxin distribution, AoB PLANTS, Volume 14, Issue 1, February 2022, plab075, https://doi.org/10.1093/aobpla/plab075

Authors' Reply: Thanks for your kind advice. We feel sorry, since it is a publication from our lab. We have corrected the statement and added related text. (Line 308-309) 

My reply:

3. Well done.

Reviewer 2 Report

- Lines 131-132 and Fig.2A: given the different focal plane between col and atann3-1 roots, authors cannot state that atann3-1 seedlings show a weakened response to eATP. Indeed, gus expression appears increased in the vasculature of atann3-1. Please comment on it.

Authors' Reply: We observed the GUS-stained sample with a general optical microscope rather than confocal laser scanning microscope, so the images indicate GUS location in the whole organ. Yes, the expression of GUS looked increased in vasculature of some atann3-1 root tips, nevertheless, in root tip cells around the QC (the area corresponding to the position shown by red arrows in Figure 2A), GUS expression did not alter markedly.

R: I do not agree with author’s response. Authors stated in the text “After ATP treatment, GUS was accumulated in root tip cells, but its abundance and the extent of its distribution were both lower than those in wild-type plants, demonstrating the weakened response of atann3–1 seedlings to eATP (Fig. 2A)”, in figure 2A increased GUS expression can be observed in the vasculature of atann3–1 mutants treated with ATP and perhaps an appreciable difference in expression in the stem cell niche is not observed.

- Fig.3B,C: in the vasculature of ATP-treated atann3-1 the DR5-GFP expression seems not present. How could authors justify it?

Authors' Reply: The DR5-GFP in vasculature of ATP-treated atann3-1 roots really exist, although a little weaker. If the image is enlarged, the fluorescence can be seen.

R: I do not agree, in the vasculature of atann3-1 treated with ATP, no DR5-GFP expression can be observed.

- lines 172-173: it is really hard to have such conclusion without a measurement of PIN2 expression in treated and untreated plants, please provide a fluorescent quantification.

line 176: to substantiate the sentence, please provide a PIN2 fluorescent quantification of atann3-1 mutants treated and untreated.

Authors' Reply: Yes, it is really hard to have such conclusion. We have revised the two sentences by deleting some words.

R: Rephrasing the text did not solve the problem. Authors should provide a quantification of PIN2, given that the reported ratio in Fig.4C is not informative on PIN2 total amount.

- To support authors statements of PIN2 and PIN3 regulation via ANN3, understand where ANN3 localize in the root and in the hypocotyl would be crucial.

Authors' Reply: Yes, it is crucial. Localization of AtANN3 in root tip cells was reported recently by Liu et al. (2021) (Reference 57). Whilst, the localization of AtANN3 in hypocotyls was not reported. We will consider your valuable advice and conduct such a detection in future work.

R: Since the authors have no information on a putative expression in the hypocotyl, they assume it, it seems premature to speak of a role of ANN3 in this region.

- The experiment reported in the paragraph “2.4. AtANN3 is involved in eATP-induced vesicle abundance in root cells” is not described in a comprehensive way. It is unclear how the authors carried this out. Please report it in detail as no materials and methods section is provided about it. Moreover, the images reported in Fig.6 are not understandable and explanatory.

Authors' Reply: In the revised part 2.4, the method is described. We also revised the figure legend to improve its understandability.

R: The problem still exists: the main text reported just some information; no methods section reported the detail of how the experiment was performed; Fig.6A was not changed.

Round 3

Reviewer 1 Report

Major issues:

1. Your revised text (lines 119-20) is an improvement but to make it even clearer I recommend changing the text further: “The ATP-suppressed root elongation was also observed in seedlings of the 2 mutant lines, and their roots were also significantly shorter than those of control mutant seedlings. Seedlings of wild-type and both mutant lines had statistically significantly the same control root lengths.”  

Reply: It has been revised following your kind advice. See line 118-120.

Response to Reply: Your new text on lines 118-120 is much clearer.

2. Although it would have been much better to image each seedling on the plate before using it for microscopy. I think adding a description to the Material and Methods section to clarify to your audience the basis of how seedlings were chosen for microscopic analyses i.e that you chose bending seedlings from both wild-type and mutant lines but mutant seedlings while showing some bending did not show as much bending as wild-type seedlings 

Reply: Thanks for your kind advice. We added some sentences to describe it in the material and method. See line 403-407.

Response to Reply: This is an improvement.

3. This experimental data needs to be included in the manuscript. However, in order to include it the authors will need to perform the key control of treating both wild-type and mutant controls and ATP-treated with and without Brefeldin A (BFA). 

Reply: We have performed the experiment and added the results in revised Figure 6. Please forgive my silly reply last time. 

Response to Reply: There are BFA-sensitive and BFA-insensitive secretory pathways in plants. The two references below support the conclusion that trafficking of PINs is BFA-sensitive. This is relevant to your data, i.e. BFA-treatment blocks the eATP-induced asymmetry of PIN2-GFP containing vesicles in wild-type and blocks the eATP-induced reduction of PIN2-GFP vesicles in the atann3 root tip cells. This new data is important and needs to be described more clearly in the Results section (see my suggestion below) AND text discussing the importance of this data needs to be added to the Discussion section (see references below that also should be added to your new text in the Discussion section). Also, importantly with the addition of this new data I noticed that your y-axis for Fig. 6B needs to be corrected. It should be changed from “amount of vesicles per cell” to amount of PIN2-GFP vesicles per cell”.

Pg 10, lines 294-299 “Added Brefeldin A (BFA) did not signifiacantly affect vesicle abundance in general, either in Col-0 or in atann3-1 seedlings, although vesicle number increased in some cells. When BFA was present in medium, eATP did not alter vesicle abundance, either in Col-0 or in atann3-1 seedlings, and the asymmetric vesicle abundance in inner side and outer side cells of the bending area was not observed (Fig. 6A and B).” 

should be changed to 

“Treatment with Brefeldin A (BFA) did not significantly affect vesicle abundance in general, either in control Col-0 or in control atann3-1 seedlings. However, BFA treatment blocked the eATP-induced asymmetrical localization of PIN2-GFP containing vesicles in wild-type root tips and blocked the eATP-induced reduction in the abundance of PIN2-GFP containing vesicles in annat3-1 root tips.”

Here are the references for new text in the Discussion section.

Nishimura T, Matano N, Morishima T, Kakinuma C, Hayashi K, Komano T, Kubo M, Hasebe M, Kasahara H, Kamiya Y, Koshiba T. 2012. Identification of IAA transport inhibitors including compounds affecting cellular PIN trafficking by two chemical screening approaches using maize coleoptile systems. Plant & Cell Physiology 53:1671–1682.

Titapiwatanakun B, Murphy AS. 2009. Post-transcriptional regulation of auxin transport proteins: cellular trafficking, protein phosphorylation, protein maturation, ubiquitination, and membrane composition. Journal of Experimental Botany 60:1093–1107.

Minor issues:

1. Text has been greatly improved but there is still a need for more proofreading. For example, pg 11, line 280 change “plant growth & development” to “plant growth and development”. Another example, on pg 3, line 79 change “AtAnn2-/ AtANN3-mediated” to “AtANN2/AtANN3-mediated”.

Reply: All revised. 

Response to Reply: This text has been corrected.

2. It was important to add this citation and reference but the new text on pg 2, lines 70-74 needs to be improved. I recommend changing “(ROS)-responsive Ca2+ or K+ channels [50–53], e.g., AtANN1 in Arabidopsis thaliana acting as a ROS-stimulated Ca2+ channel was revealed to be involved in eATP-induced Ca2+ signaling and probably also in stress-induced Ca2+ signaling [51, 54–56].” to “(ROS)-responsive Ca2+ or K+ channels [50–53]. Importantly, AtANN1 which acts as a ROS-stimulated Ca2+ channel was shown to play a major role in the eATP-induced increase in [Ca2+]cyt in Arabidopsis roots [55]. 

Reply: Revised. 

Response to Reply: This text has been corrected.

Reviewer 2 Report

The authors responded sufficiently to most of the comments provided during the last round of the peer review

Author Response

Thank you for approving the revision.